# Dental Malocclusion in Mixed Dentition Children and Its Relation to Podal System and Gait Parameters

**DOI:** 10.3390/ijerph20032716

**Published:** 2023-02-03

**Authors:** Dorota Różańska-Perlińska, Jarosław Jaszczur-Nowicki, Dariusz Kruczkowski, Joanna Magdalena Bukowska

**Affiliations:** 1Faculty of Health Sciences, Academy of Applied Medical and Social Sciences in Elbląg, 82-300 Elbląg, Poland; 2Department Physiotherapy, School of Public Health, Collegium Medicum, University of Warmia and Mazury, 10-719 Olsztyn, Poland

**Keywords:** dental malocclusion, adolescent, teenagers, gait, posture, public health

## Abstract

Background: Dental malocclusion is an increasingly frequent stomatognathic disorder in children and adolescents nowadays. The purpose of this study was to confirm or deny the correlations between body posture and malocclusion. Methods: In the study, gait, distribution of foot pressure on the ground, and body balance were examined. The research group consisted of 76 patients aged 12–15 years. The research group was obtained from patients attending periodic dental check-ups at Healthcare Center Your Health EL who agreed to participate in the study. The patients were divided into two groups without malocclusion and with malocclusion, using Angle classification, which enabled determination of the anteroposterior relationship of the first molars. The pedobarographic mat was used to analyze the distribution of foot forces on the ground, the diagnostic system Wiva^®^ Science was used for gait analysis, and Kineod 3D was used for posture analysis. The Shapiro–Wilk test used for analysis showed inconsistency with normal distribution for all measurement parameters. The Mann–Whitney U test was used for the analysis, and the significance level was set at *p* ≤ 0.05. Results: Examination of the relationship between stabilometric and gait parameters showed that the position of the mandible in relation to maxilla has an important effect on gait rhythm, gait cycle duration, and right step duration time. Patients diagnosed with malocclusion showed high-speed walking rhythm in comparison to patients with Angle’s class I (*p* = 0.010). The duration of the whole gait cycle (*p* = 0.007) and the duration of right step (*p* = 0.027) were prolonged in students without orthodontic disorders compared to the other. Conclusions: The conducted study proved that there is correlation between the presence of a stomatognathic disorder and gait cycle parameters. There is a statistically noticeable correspondence between the position of the mandible in relation to maxilla and walking rhythm, gait cycle duration, and right step duration time. Namely, students who presented malocclusion had a high-speed walking rhythm and decreased duration of the gait cycle and of the right step. On the other hand, students without disorders (Angle’s class I) showed low-speed rhythm and increased duration of the gait cycle and of the right step in comparison to Angle’s classes II and III.

## 1. Introduction

Over the last decades, the relationship between the morphological structure of the face, an integral component of the upper body, and postural system has been a constant subject of interest to health care professionals. The first research focused on potential correlations between stomatognathic apparatus and body posture was investigated by Rocabado et al. in 1982 [1]. There are various studies that revealed the significant correspondence between malocclusion and pathological orthopedic findings including not only head and neck position but also distal musculature and orientation of the limbs in equilibrium with motion and gravitation [2,3,4,5]. There are numerous examples: patients with idiopathic scoliosis perform asymmetric features of malocclusion [6], children suffering from various spinal deformities have a number of different stomatognathic disorders [7], young patients with congenital hip dislocation show risk at development of a lateral cross bite [8], and others.

Dental malocclusion is an increasingly frequent stomatognathic disorder in children and adolescents nowadays. The term malocclusion can be defined as incorrect relation between the teeth of upper and lower dental arches when they approach each other as the jaws close. Broadly speaking, malocclusion is also an imbalance between the masticatory muscle activity and temporomandibular joint function [2]. Numerous factors can cause development of malocclusion—genetic disorder, hereditary conditions, environmental elements affecting fetus [medicaments, vitamin deficiency, infectious disease, drugs and alcohol, stress, hormonal disorder], injuries, premature tooth loss, disfunctions and parafunctions of stomatognathic apparatus, and other diseases, for example rickets. Because of the multifactorial origin of malocclusion, it is difficult to analyze possible body position influence, and numerous variables should be taken into consideration [9]. According to the WHO (World Health Organization), malocclusion is the second most common oral disfunction [after cavities] in young patients. It is clear that this is a clinical issue of great importance to many health care professionals especially in treatment of malocclusion in children [10].

These connections between postural system and orthodontic disorders have generated scientific debate due to several aspects of stomatognathic system conditions that have been found to be associated with body posture [11]. Among these aspects are mandible position [12,13], dentition phase [14,15], dental [11,16] or skeletal malocclusion [17], and temporomandibular disfunctions [18,19]. Therefore, dental and especially orthodontic treatment may affect body posture in the sagittal and frontal planes, as well as influence contact between the foot and the ground, footprint, center of mass, and vice versa [5,20]. For instance, patients with retrognathic disorders change their position of the head and shoulders projecting them forward, while prognathic individuals project them opposite. Bricot explained that the position of head and shoulders is “the body’s attempt to balance against the afferent issued by temporomandibular joint” [2,21].

Postural system as well as jaw and teeth position can be examined using various criteria and methods. The most reliable method of occlusion analysis is Angle’s classification, which includes three possible ways of maxillary and mandibular canine and molar position [9,22]. Important groups of orthodontic disfunctions are lateral misalignments (cross bite), open bites—no vertical overlap or contact between the anterior or posterior teeth, and deep bites, which refer to the top teeth excessively overlapping the bottom teeth. The most common malocclusions in the European population are crossbite (36%), retromandibular disorders—class II (30%), and deep bite (19%). Rare types of malocclusion in Europe include prognathic disorders—class III (3%) and open bites (3%) [23].

There are several instruments and methods provided in the literature to control body posture: orthopedic examination of the body posture [14], clinical calculate foot posture index (FPI) or measuring the Clarke angle [24], retrospective analysis of medical documentation, and radiographs [25].

Recently, an increasing number of practitioners have used the stabilometric platform to find correlation between body posture and jaw position. Alvarez Solano et al. analyzed twelve articles on relationship between occlusion and body posture using a stabilometric platform and concluded that 66.7% of them showed correlation between body posture and dental occlusion while 33.3% found no relationship [4]. Posturography recordings were performed using a 10-Hz (Hertz) sampling frequency vertical force platform (Bio Postural System, AXA S.r.l., Vimercate [Mi], Italy) [11].

In 2021, interesting research was conducted using Cabrera-Dominguez’s team in which relevant connections between malocclusion and foot balance and body center of mass were found. This study was carried out on 409 children (222 boys and 187 girls) between 8 and 14 years old (the mean body height was 160 cm (centimeter), SD (Standard Deviation) = 8 cm, with an average body weight of 55.8 kg (kilogram), SD = 14.8 kg. Dental occlusion was assessed on the sagittal plane (Angle’s classification). The contact between the foot and the ground and the center of mass were evaluated using baropodometric measurement—the stabilometric platform, Neo-plate^®^ pressure platform. Authors found a statistically significant relationship between the plantigrade phase, the contact surface area, and center of gravity. There was a prevalence of molar and canine Angle’s class II malocclusion. In molar class II, an anterior center of gravity was predominant, in class I it was centered, and in class III it was posterior. There was significant correlation between malocclusions and the FPI of the left foot and the height of the scaphoid in the right foot [26]. The purpose of this study was to confirm or deny the correlations between body posture, considering gait parameters, distribution of foot pressure on the ground and body balance, and malocclusion.

## 2. Materials and Methods

There were 76 patients in the research group. 40 of them were male (52.6%) and 36 of them were female (47.4%). The study sample was obtained from patients who came to the Twoje Zdrowie EL Health Centre for regular dental check-ups and who agreed to participate in the study. Inclusion criteria were age between 12 and 15 years and informed consent of parents/guardians. Parents/guardians were previously informed about the study and completed a questionnaire with required participant data. The experimental research was to be carried out in 7 classes from 2 schools, with a total of 128 students between the ages of 12 and 15 years. The research data were collected from 76 patients. The rest of the students were not present at school, did not have a consent form signed by their parents, or were excluded due to the criteria mentioned above.

Exclusion criteria were: previous lower limb or upper body surgery, previous severe trauma altering the child’s initial posture, previous orthodontic and/or orthotic treatment, insufficient teeth to determine dental classification, and skeletal disorders.

The study was conducted with the consent of the Ethics Committee for Scientific Research of the University of Warmia and Mazury, Olsztyn (Decision No. 9/2018).

The orthodontic data were recorded using Angle classification, which enabled determination of the anteroposterior relationship of the jaws. This relationship can be determined for the molars and canines and classified Class I, II, and III without subdivision classes.
Class I—normal molar occlusion; the mesiobuccal cusp of the maxillary first molar aligns with the mesiobuccal groove of the mandibular first molar. The mesial incline of the maxillary canine occludes with the distal incline of the mandibular canine. The distal incline of the maxillary canine occludes with the mesial incline of the mandibular first premolar.Class II—distoocclusion; the mesiobuccal groove of the mandibular first molar is DISTALLY (posteriorly) positioned when in occlusion with the mesiobuccal cusp of the maxillary first molar. The mesial incline of the maxillary canine occludes ANTERIORLY with the distal incline of the mandibular canine. The distal surface of the mandibular canine is POSTERIOR to the mesial surface of the maxillary canine by at least the width of a premolar.Class III—mesiooclusion; the mesiobuccal cusp of the maxillary first permanent molar occludes DISTALLY (posteriorly) to the mesiobuccal groove of the mandibular first molar. Distal surface of the mandibular canines is mesial to the mesial surface of the maxillary canines by at least the width of a premolar. Mandibular incisors are in complete crossbite. Moreover, the following malocclusion features were observed: cross bites, lingual occlusion, and deep and open bites. The study data were collected by direct observation of the oral cavity [22].

### 2.1. Instruments

Weight was measured using a Tanita InnerScan^®^V model BC-545N (Tanita Corporation, Maenocho, Itabashiku, Tokyo, Japan), and height was measured using a Soehnle electronic height gauge (Soehnle, Gaildorfer Straße 6, 71522 Backnang, Germany). A pedobarographic mat was used to analyze the distribution of foot forces on the ground E.P.S R/1 (Letsens Group, Letsens S.R.L. Via Buozzi, CastelMaggiore, Bologna, Italy), diagnostic system Wiva^®^ Science (Letsens Group, Letsens S. R.L. Via Buozzi, CastelMaggiore, Bologna, Italy), and BioMech Studio software (Biomech Studio 2.0 Manual, Letsens Group, Letsens SRL Via Buozzi, CastelMaggiore, Bologna, Italy) for gait analysis. KINEOD 3D (AXS MEDICAL SAS: 3, Rue Saint-Nicolas BP 41264 F-76068 Le Havre Cedex France) was used for posture analysis.

### 2.2. Procedure

In the initial stages, head teachers and parents were consulted about the planned study. A schedule for the survey was also developed, and parental consent was obtained. A specially designed form was used for this purpose. The experimental sessions took place in the dental practice.

Natural light was used in the room. Participants were given privacy to avoid outside influences. Each person was tested in a single session so that all tasks were performed under identical conditions (ambient temperature: 22 °C). In order to collect data for the analysis of the distribution of foot forces on the ground and during gait, each participant was anonymously registered in the Biomech software (Biomech Studio 2.0 manual) with the following data: participant code, date of birth, gender, and weight and height measured by the researcher with a height measuring device while maintaining an upright posture. A Wiva^®^ Science sensor was placed on each subject at L5 vertebral level, and it sent data to a computer via a Bluetooth connection. The test procedure is shown in Figure 1.

### 2.3. Measurement Protocol

First, the participant’s occlusion was diagnosed by an orthodontic specialist. Next, the subject stripped down to his underwear and stepped onto a Tanita InnerScan^®^V model BC-545N weight-measuring device, where he stood upright with his hands along his body. In the next test, the subject would go to the E.P.S R/1 pedobarometric mat, where he stood barefoot, upright, with his arms at his body, in a natural position, without moving, looking straight ahead at a fixed point at eye level for 20 s. At the same time, patients were scanned using infrared image acquisition technology, with the KINEOD 3D posture device.

After wearing the Wiva^®^ Science sensor at the level of the L5 vertebra, the participant was tasked with walking a 15 m distance four times with a natural gait. Each time, after walking a certain distance, the subject had to stop at the instructor’s signal, turn around, wait for the signal again, and then walk another 15 m distance. The measurement values of gait parameters test were: rhythm [steps/min], duration of gait cycle [s], support duration [% gait cycle], swing duration [% gait cycle], double support time [% gait cycle], single support time [% gait cycle], left step length [%], right step length [%], left step duration [%], right step duration [%], left support duration [% gait cycle], right support duration [% gait cycle], left swing duration [% gait cycle], and right swing duration [% gait cycle].

### 2.4. Statistical Analysis

In the gait study, it can be assumed that the distribution of data does not follow a normal distribution, so it was decided to verify the appropriate statistical procedure for the sample size. The Shapiro–Wilk test used for analysis showed inconsistency with normal distribution for all measurement parameters. Therefore, the statistical non-parametric method (Mann–Whitney U test) was used in further analysis. The measures used to characterize descriptive statistics were the median and the dispersion of measurement values (quartiles). The level of significance in the study was set at *p* < 0.05. Statistical analyses were performed using Statistica (StatSoft Poland, Krakow, Poland, version 13.3), and to create figures SPSS 28.0 (IBM Corporation, Armonk, NY, USA) software was used.

## 3. Results

Applying basic statistical analysis, based on reference to population norms, no significant abnormalities in the basic gait parameters of the individuals in the study population were noticed, and those that deviated from the range of the norm were marginal (Table 1).

Step rhythm abnormalities were few, about 6% of the study population, where the rhythm was accelerated (four individuals) and about 12% with a slowed rhythm (eight individuals). The mean value was 53.50% and was within the normal range for both the girls’ and boys’ populations. There were also few abnormalities in the duration of the entire gait cycle, where a prolonged cycle was noted in about 10% (seven individuals) and a shortened cycle in 6% (four individuals). The average value was 1.12% and was within the normal range for both the girls’ and boys’ populations. Abnormalities in the duration of prolonged support were noted in about 17% of the subjects (11 individuals) and shortened time in 10% (seven individuals). The mean value was 60.80% and was within the norm for both the girls’ and boys’ populations. The highest deviation values were related to the parameter of transfer time. Values of shortened transfer time were recorded in 45% of the subjects (30 individuals), while prolonged transfer time was found in 3% of the subjects (two individuals). The average value was 37.2% and was within the norm for both the girls’ and boys’ populations.

By subjecting the basic statistical analysis of the values of the podiatric examination parameters, it was noted that the majority of the examined individuals had excessive foot pronation (Table 2). In the analysis, increased load on the right foot relative to the left foot was of particular note. Excessive arches of the left foot (14%) were noted in 38 individuals (58%), while the right foot was noted in 43 (65%). At the same time, significant flattening of the left foot (above 35%) was noted in one individual and the right foot in two subjects. The overall load for the right and left foot was different in the study population. A higher load on the left foot (above 50%) was recorded in 47 subjects (71%), while the right foot was recorded in 19 (29%). The average load on the right foot was 53.65%, while the left foot was 46.90%. The analysis of the results of the stabilographic test, toward the assessment of the ability to maintain body balance, did not reveal any significant asymmetry in the mean values (Table 2).

The evaluation of the correlation between stabilographic, podiatric examination, and gait parameters was positively verified only for selected gait parameters. In Table 3, statistical significance was noted for key gait parameters such as rhythm (*p* = 0.010) and gait cycle duration (*p* = 0.007) (Figure 2 and Figure 3). In addition, such a relationship was found for the duration of the right step (*p* = 0.027), a parameter that remains highly correlated with the duration of the whole cycle.

The recorded directly proportional relationship for the gait rhythm parameter indicates an increase in the occurrence of malocclusion in individuals with high step frequency. The consequence is an inversely proportional dependence of the duration of the entire cyclic walk (Figure 2 and Figure 3).

## 4. Discussion

The purpose of this study was to confirm or deny the correlations between body posture and malocclusion. Many authors analyzed the problem and found that there is relationship between body equilibrium and jaw position. Therefore, dental practitioners assume that orthodontic treatment may significantly affect the position of the spine as well as lower limbs [14].

As a matter of fact, some practitioners have used stabilometric platforms to find correlations between body posture and jaw position, but in this study researchers used newer and more advanced instruments: a pedobarographic mat to analyze the distribution of foot forces on the ground and Kineod 3D for posture analysis. In addition, no studies on the subject could be found in the available literature on the relationship between dental malocclusion and gait parameters, which is why the diagnostic system Wiva^®^ Science was used for gait analysis.

The study identified a greater prevalence of Angle’s class II in examined patients. Collected data included information about body posture (spine position), body equilibrium, foot balance, and gait rhythm. Although other researchers found correspondence between malocclusion and body position and equilibrium in their studies [4,26], no statistically significant relationship was discovered during this examination. This may be explained by the fact that the group of patients was limited to small population or that there was no analysis of body posture differentiating types of malocclusions, because 32 of 76 students did not show any type of stomatognathic disorder.

However, examination of the relationship between stabilometric and gait parameters showed that the position of the mandible in relation to maxilla has important effect on gait rhythm, gait cycle duration, and right step duration time. Patients diagnosed with malocclusion showed high-speed walking rhythm in comparison to patients with Angle’s class I. Moreover, the duration of the whole gait cycle and the duration of right step were prolonged in students without orthodontic disorders compared to the other. The relationship between stabilometric and gait parameters was also demonstrated in the study by Fujimoto et al. [27]. Thus, it may be considered that the change of mandibular position in malocclusion caused different positioning of head and neck muscles, thereby impairing body coordination leading to changes in walking balance. Furthermore, imbalanced position of the jaws may be believed to generate modifications in the location of the center of body gravity. It may be assumed that malocclusion patients attempt to compensate the disturbance of equilibrium by taking larger numbers of more rapid steps, which is statistically noticeable in the duration time of right step (presumably the right foot was dominant, but it would demand additional inquiry). On the other hand, high-speed walking rhythm and shortened gait cycle may result from the fact that malocclusion patients have decreased body stability, and when they raise the foot, their body equilibrium is affected and when they are unable to catch their balance, they are forced to quickly put their foot to the ground again. Certainly, the subject demands more examination and selection of more specific criteria in order to explore more correlations between the body posture and stomatognathic apparatus disorders and to broaden knowledge about possible methods for malocclusion and posture defect prevention. A study by Bardellini et al. showed that treating malocclusion has a positive effect on children’s posture and balance [28].

Studies by Jaszczur-Nowicki et al. [29] and Bukowska et al. [30,31] show that an ex-ternal load in the form of a 5 kg backpack and specifics of physical effort affect the distribution of foot forces on the ground and body balance. Prolonged external loading can contribute to postural defects. It can be assumed by analyzing the results of the above works and the results of our own research that postural defects may contribute to the aggravation of malocclusion, to which, among other things, too much external load may contribute.

At this point, the strengths of the study should be indicated—using new and more advanced instruments to analyze the distribution of foot forces on the ground and posture analysis and founding relations between dental malocclusions and gait parameters. There are also limitations of the study that should be mentioned: the small population or nonclassification of the sagittal skeleton of patients and no postural analysis to differentiate types of malocclusion.

## 5. Conclusions

(1) This study proved that there is correlation between presence of stomatognathic disorder and gait cycle parameters.

Proposals for further studies:

(2) The type of malocclusion might be taken into consideration while analyzing obtained gait data (students might be divided into groups of three Angle’s classes, not only a group with Angle’s I class and a group having different malocclusions).

(3) The dominant side of the children examined should be taken into account.

(4) Students’ body composition and physical parameters (height, weight) might be considered.

(5) From a therapeutic point of view, it is necessary for orthodontists and physiotherapists to work together to diagnose and treat malocclusion in order to develop the most effective therapy.

## Figures and Tables

**Figure 1 ijerph-20-02716-f001:**
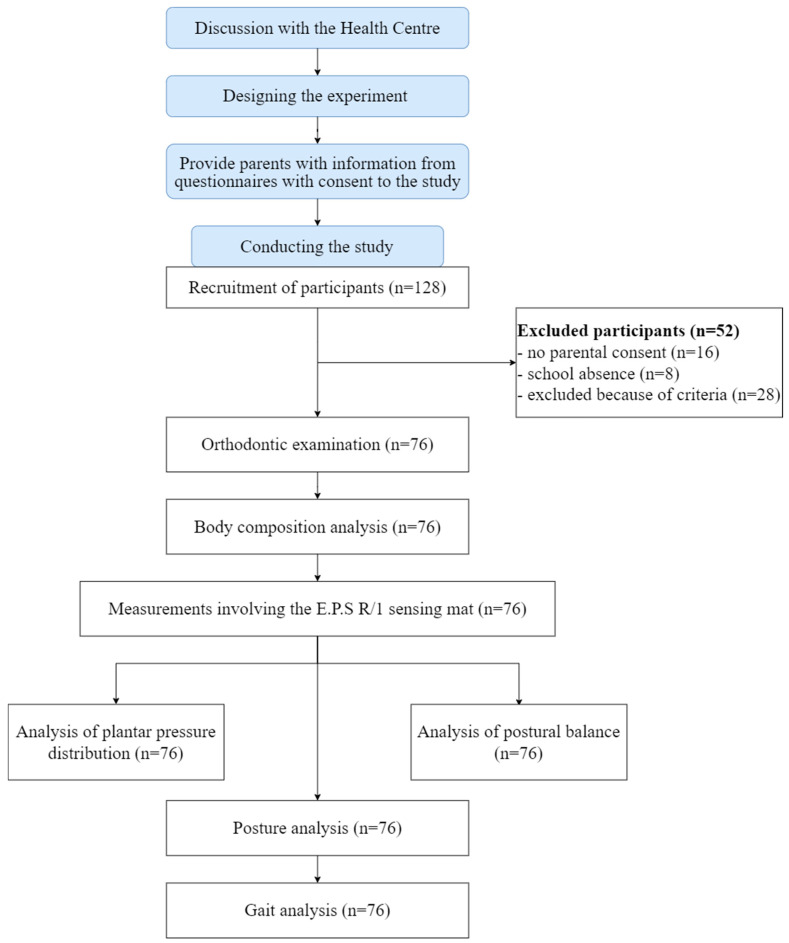
Scheme of the test procedure.

**Figure 2 ijerph-20-02716-f002:**
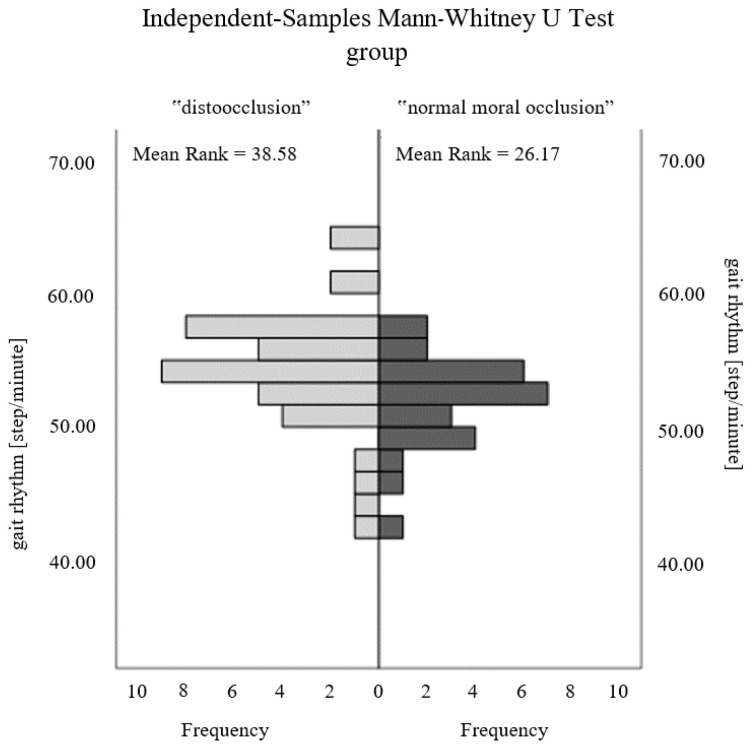
Comparison of the gait rhythm parameter between the study groups.

**Figure 3 ijerph-20-02716-f003:**
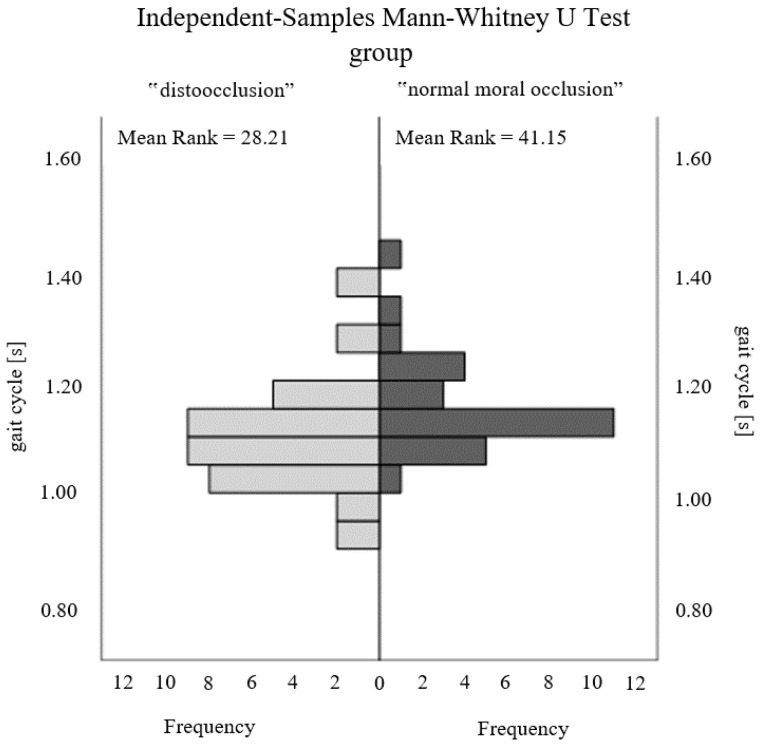
Comparison of the gait cycle parameter between the study groups.

**Table 1 ijerph-20-02716-t001:** Fundamental statistics of measurement values of gait parameters test.

Parameters	Rhythm [Steps/min]	Duration of Gait Cycle [s]	Support Duration [% Gait Cycle]	Swing Duration [% Gait Cycle]	Double Support Time [% Gait Cycle]	Single Support Time [% Gait Cycle]	Left Step Length [(%]	Right Step Length [%]	Left Step Duration [%]	Right Step Duration [%]	Left Support Duration [% Gait Cycle]	Right Support Duration [% Gait Cycle]	Left Swing Duration [% Gait Cycle]	Right Swing Duration [% Gait Cycle]
M	53.50	1.12	60.80	37.20	11.75	37.20	50.45	49.55	50.05	50.05	60.80	60.90	37.85	37.15
Q_1_	50.53	1.07	59.53	36.05	10.70	36.05	48.05	48.00	49.20	49.20	59.10	59.63	36.33	35.43
Q_3_	55.98	1.19	61.95	38.40	12.78	38.40	52.00	51.95	50.80	50.88	61.70	62.18	39.08	38.35

M—Median, Q1—the first quartile, Q3—the third quartile.

**Table 2 ijerph-20-02716-t002:** Fundamental statistics of measurement values of podiatric and stabilographic examination parameters.

Parameters	Podology								Stabilographic
	Left Foot Load [%]	Right Foot Load [%]	Left Foot			Right Foot			The Total Surface Area of the Body’s Center of Gravity [mm^2^]
			Forefoot [%]	Metatarsus [%]	Heel [%]	Forefoot [%]	Metatarsus [%]	Heel [%]	
M	53.65	46.90	42.95	8.95	45.30	46.40	8.30	38.95	111.74
Q_1_	48.38	41.93	35.35	0.38	37.25	34.80	2.00	32.50	34.61
Q_3_	58.25	52.08	50.38	20.70	51.88	57.55	20.10	53.70	190.61

M—Median. Q1—the first quartile. Q3—the third quartile.

**Table 3 ijerph-20-02716-t003:** Relationship of gait parameters with mandibular alignment (Mann–Whitney U test for *p* ≤ 0.05).

Parameter	Z	*p*
Rhythm [steps/min]	2.576	0.010 *
Duration of gait cycle [s]	−2.687	0.007 *
Support duration [% gait cycle]	0.685	0.494
Swing duration [% gait cycle]	−1.435	0.151
Double support time [% gait cycle]	1.285	0.199
Single support time [% gait cycle]	−1.532	0.125
Left step length [%]	1.487	0.137
Right step length [%]	−1.467	0.142
Left step duration: [%]	1.806	0.071
Right step duration: [%]	−2.217	0.027 *
Left support duration [% gait cycle]	1.226	0.220
Right support duration [% gait cycle]	0.078	0.938
Left swing duration [% gait cycle]	−1.311	0.190
Right swing duration [% gait cycle]	−1.239	0.215

Z: U Mann–Whitney test. *p*: significance level. * *p* ≤ 0.05.

## Data Availability

Not applicable.

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
