# Peer review of "Dental Malocclusion in Mixed Dentition Children and Its Relation to Podal System and Gait Parameters"

_ijerph, 2023, doi:10.3390/ijerph20032716_

Round 1

Reviewer 1 Report

Dental maloclussion in mixed dentition children and its relation to podal system and gait parameters

Reviewer Report

The subject of study is interesting but notable on an important subject. But overall major revision is required. I think it will be ready for publication once these corrections are made:

ABSTRACT

The abstract is the essence of the work. When the abstract is read, the whole of the study should be understood. Therefore, I request that it be corrected by taking into account what I have suggested below.

Lines 13-14: ‘In the study, gait, distribution of foot pressure on the ground and body balance were examined.’ instead of ‘The study examined gait, distribution of 13 foot pressure on the ground and body balance.’, in my opinion

Line 14: repetitive words, please remove

Line 14-17: ‘Research group or Study sample’ or ‘Patients or subjects’. It can be fluent if continued in the same terminology for clarity

Line 16-18: ‘Angle classification' is the classification of molar relation’. The expression ‘anteroposterior relationship of the jaws’ is skeletally attributable. Please simplify and clarify.

Did you classify molar relations as 1,2,3 only, or with subdivision classes as well? Please specify.

Also, what skeletal class were these patients in? Which of 1, 2, 3 did you choose? please specify.

Methods: Since an important issue in the field has been handled from a different perspective, the investigated parameters need to be explained a little more.

Statistical analyzes used and significance value considered should be added.

Results: Results should be presented with statistical significance.

Conclusion: It should provide information that has been reached from the results of the study. It should be developed by making a suggestion or general comment and presenting expressions.

Line 25: The expression 'Submitted study' is not appropriate. Please change it.

INTRODUCTION

The introduction is generally well written.

Line 53: ‘WHO’ should be used after the abbreviation is explained where it first appears.

Lines 86,90,91: ‘Hz’, ‘cm’, ‘SD’, ‘kg’ abbreviations should be used after they are explained where they first appear.

MATERIALS AND METHODS

Lines 121-123: What has been thought about subgroups of molar relations? Only those of double-sided class 1,2,3 were included? please specify. Information should also be given about the sagittal skeletal classes of the patients.

Lines 124-141: This should be supported by reference.

Line 184: ‘Statistical Analysis’ instead of ‘Data Analysis’

Lines 191-193: Did you use Statistica or SPSS? What was done in which, please specify?

Wouldn’t it be more correct to choose Kolmogorov-Simirnow instead of Shapiro-Wilk since the sample size was more than 50? This is just suggestion.

RESULTS

All tables should be edited in accordance with the journal template.

Explanations of the abbreviations in the table should be written under Table 1.

Use dots instead of commas in tables.

The gait parameters mentioned in Table 1 should also be mentioned in the material method section.

Explanations of the abbreviations in the table should be written under Table 2.

Explanations of the abbreviations in the table should be written under Table 3.

In Table 3: Those found to be statistically significant should be indicated in the table with asterix instead of red color. And the statistical analyzes used and significances should be explained with abbreviations under the table. (For example: N: sample size, a: Mann–Whitney U test, p: significance level, * p < 0.05.)

Line 251: I think figures 2 and 3 should be interpreted with a little more explanation.

The relationships between malocclusions and investigated parameters should be presented as table/s.Was not a table statistically comparing malocclusion and gait parameters not the main scope of this study?

DISCUSSION

Discussion section is too brief. The discussion should be further developed.

It would be good to add a paragraph explaining which scientific deficiencies this study is planned to address in the literature in the field.

A paragraph describing the strengths and limitations of the study should be added.

Line 264,265: This expression must be supported by reference/s.

CONCLUSIONS

As a conclusion of a study, it is too long written.

The conclusion is not a repetition of the results. Some data related to results should be written in the discussion section, and its relationship with the literature should be compared.

It should present the conclusions of the study and further studies proposal in a simple and understandable manner.

REFERENCES

References should be prepared in accordance with the template.

In the references, it is seen that some journal names are written short and some are long. Please correct them all according to the template.

Author Response

Dear Recenzent, thank you very much for your very accurate and useful comments and suggestions. We have made these corrections. All changes are marked in yellow in the text.

Reviewer Report

The subject of study is interesting but notable on an important subject. But overall major revision is required. I think it will be ready for publication once these corrections are made:

ABSTRACT

The abstract is the essence of the work. When the abstract is read, the whole of the study should be understood. Therefore, I request that it be corrected by taking into account what I have suggested below.

Lines 13-14: ‘In the study, gait, distribution of foot pressure on the ground and body balance were examined.’ instead of ‘The study examined gait, distribution of 13 foot pressure on the ground and body balance.’, in my opinion

Reply: Thank you very much for your suggestion, we replaced the sentence: ‘The study examined gait, distribution of foot pressure on the ground and body balance.’ with a sentence: ‘In the study, gait, distribution of foot pressure on the ground and body balance were examined.’

Line 14: repetitive words, please remove

 Reply: We removed repetitive words.

Line 14-17: ‘Research group or Study sample’ or ‘Patients or subjects’. It can be fluent if continued in the same terminology for clarity

Reply: Thank you very much for your suggestion, we used only ‘research group’ and ‘patients’.

Line 16-18: ‘Angle classification' is the classification of molar relation’. The expression ‘anteroposterior relationship of the jaws’ is skeletally attributable. Please simplify and clarify.

Did you classify molar relations as 1,2,3 only, or with subdivision classes as well? Please specify.

Reply: We replaced the sentence: ‘anteroposterior relationship of the jaws’ with a sentence ‘anteroposterior relationship of the first molars’.

Also, what skeletal class were these patients in? Which of 1, 2, 3 did you choose? please specify.

Reply: We classified only 1,2,3 molar relations without subdivision classes. We added these information into the text. Skeletal class we counted as the exclusion criteria. We added these information into the text.

Methods: Since an important issue in the field has been handled from a different perspective, the investigated parameters need to be explained a little more.

Statistical analyzes used and significance value considered should be added.

Reply: Added a little more information about investigated instruments and statistical analyzes.

Results: Results should be presented with statistical significance.

Reply: Results presented with statistical significance.

Conclusion: It should provide information that has been reached from the results of the study. It should be developed by making a suggestion or general comment and presenting expressions.

Reply: Supplemented with additional suggestions and expressions.

Line 25: The expression 'Submitted study' is not appropriate. Please change it.        

Reply: Thank you very much for your suggestion. The expression 'Submitted study' was instead into ‘Conducted study’.

INTRODUCTION

The introduction is generally well written.

Line 53: ‘WHO’ should be used after the abbreviation is explained where it first appears.

Reply: Thank you very much for your suggestion. Abbreviations are explained where it first appears.

Lines 86,90,91: ‘Hz’, ‘cm’, ‘SD’, ‘kg’ abbreviations should be used after they are explained where they first appear.

Reply: Thank you very much for your suggestion. Abbreviations are explained where it first appears.

MATERIALS AND METHODS

Lines 121-123: What has been thought about subgroups of molar relations? Only those of double-sided class 1,2,3 were included? please specify. Information should also be given about the sagittal skeletal classes of the patients.

Reply: We classified only 1,2,3 molar relations without subdivision classes. We added the information into the text. Skeletal class we counted as the exclusion criteria. We added the information into the text.

Lines 124-141: This should be supported by reference.

Reply: Added the reference

Line 184: ‘Statistical Analysis’ instead of ‘Data Analysis’

Reply: Thank you very much for your suggestion, ‘Statistical Analysis’ was instead of ‘Data Analysis’

Lines 191-193: Did you use Statistica or SPSS? What was done in which, please specify?

Reply: Statistica program was used to analyze the data, then to create graphs the data was analyzed in SPSS. The results obtained were the same in both programs, so it was decided to create graphs in SPSS. The information is included in the text.

Wouldn’t it be more correct to choose Kolmogorov-Simirnow instead of Shapiro-Wilk since the sample size was more than 50? This is just suggestion.

Reply: We agree that for samples larger than 50 it is recommended to use the Kolmogorov-Simirnov test, but computer simulations (Razali, Yap, 2011 and Yap, Sim, 2011) indicate that the Shapiro-Wilk test is the strongest test for all distribution types and sizes, and that it can be used for samples in the range of 3 < n <5000.

Razali, M., Yap, B. (2011). Power Comparisons of Shapiro-Wilk, Kolmogorov-Smirnov, Lilliefors and Anderson-Darling Tests. J. Stat. Model. Analytics. 2.

 Yap, B. W. & Sim, C. H. (2011) Comparisons of various types of normality tests, Journal of Statistical Computation and Simulation, 81:12, 2141-2155, DOI: 10.1080/00949655.2010.520163.

RESULTS

All tables should be edited in.

Reply: The tables have been edited in accordance with the journal template

Explanations of the abbreviations in the table should be written under Table 1.

Reply: The abbreviations used in the table are written under Table 1.

Use dots instead of commas in tables.

Reply: Thank you for your suggestion. The commas have been changed to periods in the tables.

mentioned in Table 1 should also be mentioned in the material method section.

Reply: The gait parameters have been mentioned in the material method section.

Explanations of the abbreviations in the table should be written under Table 2.

Reply: The abbreviations used in the table are written under Table 2.

Explanations of the abbreviations in the table should be written under Table 3.

In Table 3: Those found to be statistically significant should be indicated in the table with asterix instead of red color. And the statistical analyzes used and significances should be explained with abbreviations under the table. (For example: N: sample size, a: Mann–Whitney U test, p: significance level, * p < 0.05.)

Reply: The abbreviations used in the table are written under Table 3.

Line 251: I think figures 2 and 3 should be interpreted with a little more explanation.

The relationships between malocclusions and investigated parameters should be presented as table/s.Was not a table statistically comparing malocclusion and gait parameters not the main scope of this study?

Reply: Thank you for your suggestion. We have added a more detailed description of the graphs. The relationship between malocclusion and the parameters studied are included in Table 3.

DISCUSSION

Discussion section is too brief. The discussion should be further developed.

It would be good to add a paragraph explaining which scientific deficiencies this study is planned to address in the literature in the field.

A paragraph describing the strengths and limitations of the study should be added.

Reply: We added the paragraph explaining which scientific deficiencies this study is planned to address in the literature in the field and describing the strengths and limitations of the study.

Line 264,265: This expression must be supported by reference/s.           

Reply: Added the reference.

CONCLUSIONS

As a conclusion of a study, it is too long written.

The conclusion is not a repetition of the results. Some data related to results should be written in the discussion section, and its relationship with the literature should be compared.

It should present the conclusions of the study and further studies proposal in a simple and understandable manner.

Reply: Thank you for your suggestion. We have corrected the conclusion.

REFERENCES

References should be prepared in accordance with the template.

In the references, it is seen that some journal names are written short and some are long. Please correct them all according to the template.

Reply: The names of the journals in the references have been corrected according to the template.

Reviewer 2 Report

Dear Authors here are my recomandation

Author Response

Dear Recenzent, thank you very much for your very accurate and useful comments and suggestions. We have made these corrections. All changes are marked in yellow in the text.

Capter 2. Materials and Methods  

121-141 

The description of Angle classes I,II and III are not personally discovers of the Authors and please must be cited the source.

Reply: Added the reference.

Capter 3. Results

 214-216  Table 1 

244-246 Table 2

 The gain Parameters  M, Q1 and Q3 are not explained in the text.

Reply: The parameters M, Q1 and Q3 are explained in the text.

Please explain the acronym or abbreviation what it is  with  * in description of the Tables with corresponding information in text.

Reply: Thank you for your suggestion, acronyms and abbreviations have been explained under the tables.

257  Table 3 

What represent the acronym, abbreviation or letter  U, Z and P ?

Reply: Thank you for mentioning this mistake. U represents the value of the Mann-Whitney U test used for small sample sizes <20. Since our group size was 76, we decided to remove the U column from Table 3. Z represents the value of the Mann-Whitney U test used when the sample size is greater than 20. P - the level of statistical significance. In the table, we changed the P designation to p.

Please explain what U, Z, P are using * in description of the Table with corresponding information in text.

Reply: The letters used in Table 3 are explained below the table.

Please explain why a part of the values from P and U Colum are marked with red color

Reply: The red color indicated statistically significant values. This has been changed for editorial requirements. Statistically significant values are marked with *.

In Figure 2 and Figure 3, the related text next to the graphic (yellow) is not well readable, so please replace the text with a more readable font without blurring shape.

Reply: Thank you for your suggestion. The font in the figures has been changed.

Capter 5. Conclusion

Please share your opinion on what should be done from a therapeutic point of view?

More sport to have a correct posture and thus a normal development of the stomatognathic system or precise indications of more sport during the orthodontic treatment?

What is your opinion after the exposed study?

Reply: Our opinion has been added in conclusion.

Reviewer 3 Report

Dear colleagues!

Your study is interesting, but not a new approach to evaluating malocclusion and podal system dependency.

You know the works https://www.frontiersin.org/articles/10.3389/fped.2021.654229/full, https://www.mdpi.com/2227-9032/8/4/485 and https://journals.lww .com/md-journal/Fulltext/2018/05110/Relationship_between_foot_posture_and_dental.44.aspx which are, in many ways, more evidence than your limited study.

I would like to draw attention to the definition of sample size in your work, as well as relevance and discussion. Taking into account the above works, you should form a null hypothesis more transparently and resubmit the article for consideration.

Author Response

Dear Recenzent, thank You for Your very accurate and useful comments and suggestions.

Dear colleagues!

Your study is interesting, but not a new approach to evaluating malocclusion and podal system dependency.

You know the works https://www.frontiersin.org/articles/10.3389/fped.2021.654229/full, https://www.mdpi.com/2227-9032/8/4/485 and https://journals.lww .com/md-journal/Fulltext/2018/05110/Relationship_between_foot_posture_and_dental.44.aspx which are, in many ways, more evidence than your limited study.

I would like to draw attention to the definition of sample size in your work, as well as relevance and discussion. Taking into account the above works, you should form a null hypothesis more transparently and resubmit the article for consideration.

Reply: Regarding the relevance of the results obtained, thank you for pointing out the problem of sampling. The measurements we carried out were designed to obtain a sample of no less than 100 respondents. This was done in accordance with sampling for a limited (finite) population. As a result, those excluded managed to obtain results for 76 respondents. In our team discussion we decided to carry out an analysis of the collected material. We are aware that our observations and conclusions will be characterised by a certain degree of certainty and will not be subject to any significant error of accuracy. At the same time, we assure you that we always approach the issue of sample size with particular care.

Thank you for your guidance on the null hypothesis. However, we have not seen such a hypothesis explicitly stated in the articles cited and suggest that we include a clarified aim for our study.

The discussion was supplemented by suggestions from other reviewers.

Round 2

Reviewer 1 Report

Dental maloclussion in mixed dentition children and its relation to podal system and gait parameters

Reviewer Report Round 2

I think the study has improved considerably. And I thank to the authors for that. However, I think some minor revisions would make the study much better.

ABSTRACT

It is insufficient to simply specify the normality distribution. The nonparametric tests used and the considered statistical significance value should also be mentioned.

MATERIALS AND METHODS

I understand and accept your add of ‘skeletal disorders’ in the exclusion criteria of the study.

What I mean is what is the sagittal skeletal class of patients (Skeletal class 1, 2, or 3).

If this has been neglected, this needs to be stated. Because the results of the research are more likely to be affected by the sagittal skeletal relationships of the patients rather than the sagittal dental relationships. And this should be stated in the limitations of the study.

RESULTS

Lines 255:  p < 0.05 or p ≤ 0.05 ? Please choose one.

Lines 255:  Writings must be written in English.

Lines 321:  ‘This’ instead of ‘Submitted’, in my opinion.

Author Response

Dear Reviewer,

We are pleased that you appreciate the changes we have made. Thank you for taking the time to review our article again and for all your comments and suggestions. Please see the attachment.

Reviewer 2 Report

Good luck with the future research

Author Response

We are pleased that you appreciate the changes we have made. Thank you for taking the time to review our article.

Reviewer 3 Report

Dear colleagues! Your corrections are accepted. I am satisfied with the answers

Author Response

(The authors gave the same response as above.)
